# Forget Less, Retain More: A Lightweight Regularizer for Rehearsal-Based Continual Learning

**Lama Alssum**[*]   **Hasan Abed Al Kader Hammoud**   **Motasem Alfarra**   **Juan C Leon Alcazar**
**Bernard Ghanem**
*King Abdullah University of Science and Technology (KAUST)*
[*]*Corresponding author: lama.alssum.1@kaust.edu.sa*

**Reviewed on OpenReview:** *https://openreview.net/forum?id=CJw1ZjkJMG&noteId=uvFvIOQDES*

## Abstract

Deep neural networks suffer from catastrophic forgetting, where performance on previous tasks degrades after training on a new task. This issue arises due to the model's tendency to overwrite previously acquired knowledge with new information. We present a novel approach to address this challenge, focusing on the intersection of memory-based methods and regularization approaches. We formulate a regularization strategy, termed Information Maximization (IM) regularizer, for memory-based continual learning methods, which is based exclusively on the expected label distribution, thus making it class-agnostic. As a consequence, IM regularizer can be directly integrated into various rehearsal-based continual learning methods, reducing forgetting and favoring faster convergence. Our empirical validation shows that, across datasets and regardless of the number of tasks, our proposed regularization strategy consistently improves baseline performance at the expense of a minimal computational overhead. The lightweight nature of IM ensures that it remains a practical and scalable solution, making it applicable to real-world continual learning scenarios where efficiency is paramount. Finally, we demonstrate the data-agnostic nature of our regularizer by applying it to video data, which presents additional challenges due to its temporal structure and higher memory requirements. Despite the significant domain gap, our experiments show that IM regularizer also improves the performance of video continual learning methods.

## 1 Introduction

Continual learning (CL) aims to develop models that can learn from evolving data distributions with minimal forgetting Rolnick et al. (2019). Due to the high computational and financial costs of training deep neural network models and growing concerns over privacy regulations, the applicability of CL in various real-world scenarios has become increasingly critical. For instance, video-sharing platforms such as YouTube and TikTok receive millions of newly uploaded videos daily, each introducing new trends, visual concepts, and styles. In these dynamic environments, traditional training algorithms for deep learning models struggle to maintain their performance due to the necessity of frequent retraining, which is resource-intensive and impractical at scale. CL can significantly enhance the effectiveness of models designed for such dynamic data streams by continually adapting previously trained models, rather than retraining them from scratch as new data is made available.

In recent years, memory-based methods Rolnick et al. (2019); Buzzega et al. (2020b), also known as rehearsal methods, have emerged as the front-runners in CL, demonstrating better performance at mitigating forgetting compared to their regularization-based counterparts. This superior performance of rehearsal methods is attributed to the use of a memory buffer, a dedicated storage that retains a subset of training data from previously learned tasks. By having access to a subset of past samples, the model can estimate class prototypes effectively thus alleviate forgetting despite distribution shifts Rolnick et al. (2019); Rebuffi et al.

(2017). This ability to retain past information enables rehearsal-based approaches to maintain stability in long-term learning while adapting to new tasks.

The performance improvement of rehearsal methods over regularization methods comes at the cost of increased memory requirements and greater computational overhead. Experience replay methods retrain for several epochs on both the current data and memory buffer samples, effectively approximating a joint distribution whenever new data becomes available. This continuous reprocessing of stored data samples not only increases computational demands but also increases training time, making it less scalable for large datasets. This computational penalty is further emphasized outside the image domain; for example, in video data, a single minute-long video recorded at 30 frames per second occupies as much memory as 1800 individual images. Despite consuming a significant storage, such video typicality represents a single instance of a class within the memory buffer, thus limiting the diversity of stored information and further challenging the efficiency of experience replay in temporal data.

In this paper, we introduce a class-agnostic regularization strategy for continual learning (CL), termed Information Maximization (IM), which operates over the distribution of network predictions Liang et al. (2020). IM is lightweight, efficient, and orthogonal to key design choices in CL algorithms, as it works across different rehearsal-based methods, memory budgets, and number of tasks. IM jointly encourages prediction diversity and confidence, resulting in robust feature representations under distribution shifts, thereby improving generalization to past tasks while minimizing memory and computational overhead. Extensive empirical evaluation shows that IM emerges as an effective regularization technique, outperforming current regularization strategies tailored for the CL setting in both accuracy and retention of past knowledge. Moreover, IM is not specific to the image CL domain; we further validate its effectiveness by applying it to a video continual learning setup, where it demonstrates similarly improved results in handling the challenges posed by temporal dependencies and increased data complexity.

**Contributions.** Our work makes two main contributions: **(i)** We provide a systematic evaluation of four major regularization techniques: Elastic Weight Consolidation (EWC), Synaptic Intelligence (SI), Entropy Minimization (EM), and Information Maximization (IM), applied to image continual learning. This evaluation highlights the advantages of the proposed IM regularizer, demonstrating its superiority in terms of both performance and overall reduction in catastrophic forgetting. Furthermore, we demonstrate that IM is orthogonal to critical design choices in CL (rehearsal-based method, memory size, and number of tasks), enabling straightforward integration while preserving computational efficiency. **(ii)** We extend our analysis beyond image-based settings by demonstrating the applicability of IM within the context of video continual learning. Given the additional complexity of temporal dependencies and larger data volumes in videos, our results show that IM maintains its effectiveness, achieving substantial gains over traditional memory-based baselines while preserving computational efficiency.

## 2 Related Work

**Image Continual Learning.** In the field of image-based continual learning, numerous innovative approaches have been proposed to address catastrophic forgetting. Memory-based methods, such as iCaRL Rebuffi et al. (2017), utilize incremental classifiers and representation learning to balance new and old knowledge, while GEM Lopez-Paz & Ranzato (2017) and its more efficient variant A-GEM Chaudhry et al. (2019a) optimize gradient-based episodic memory to mitigate catastrophic forgetting. Other approaches, including DER Buzzega et al. (2020b), enhance rehearsal by incorporating logit distillation, while CoPE De Lange & Tuytelaars (2021) leverages class prototypes to structure the latent space, and ER-ACE Caccia et al. (2021) modifies cross-entropy loss to address task imbalance. Recent work includes Refresh Learning Wang et al. (2024), which unifies rehearsal with selective unlearning to refresh model knowledge, and STAR Eskandar et al. (2025), a plug-and-play regularizer that leverages stability-inducing weight perturbations during rehearsal to mitigate forgetting. Regularization-based methods aim to preserve past knowledge by constraining weight updates, typically by identifying the importance of parameters, like Elastic Weight Consolidation (EWC) Kirkpatrick et al. (2017) and Synaptic Intelligence (SI) Zenke et al. (2017). Architectural innovations also play a crucial role in continual learning, with L2P Wang et al. (2022c) demonstrating the effectiveness of learnable prompts in guiding pre-trained models without relying on a rehearsal buffer, and

DualPrompt Wang et al. (2022b) introducing a two-level prompting mechanism for transformer-based architectures. These diverse approaches underscore the rapid advancements in continual learning, paving the way for more scalable and adaptable models in real-world applications.

**Video Continual Learning.** To mitigate catastrophic forgetting in video data, various strategies have been developed, which can be broadly categorized into regularization and memory-based techniques. While regularization methods apply constraints to preserve previous knowledge, memory-based approaches leverage data or representations from past tasks. When analyzing video continual learning, the importance of memory becomes even more pronounced due to the temporal complexity and higher dimensionality of video data. SMILE Alssum et al. (2023) underscores this by proposing an efficient replay mechanism that stores a single frame per video, emphasizing video diversity over temporal information. This approach addresses memory constraints effectively, showcasing the critical role of memory in video continual learning. vCLIMB Villa et al. (2022) and PIVOT Villa et al. (2023) introduce novel benchmarks and methods focusing on class incremental learning and the use of prompting mechanisms, respectively, pushing the boundaries of current methodologies. Utilizing Winning Subnetworks for efficient learning Kang et al. (2023), and creating multi-modal datasets for egocentric activity recognition Xu et al. (2023) illustrate the expanding scope of continual learning in video domains. Additionally, Continual Predictive Learning Chen et al. (2022) and approaches to Video Object Segmentation as a continual learning task Nazemi et al. (2023) represent significant advancements in handling non-stationary environments and long video sequences. Finally, efforts to learn new class representations while preserving old ones through time-channel importance maps Park et al. (2021) further demonstrate the innovative and diverse strategies being developed for video continual learning.

**Test-Time Adaptation.** Test-Time Adaptation (TTA) aims to alleviate performance drop of pretrained models at test time when exposed to domain shifts Sun et al. (2020); Alfarra et al. (2023). Earlier works augmented the training objective with a self-supervised loss function that is later leveraged at test time to combat domain shifts Sun et al. (2020); Liu et al. (2021). More recent TTA methods optimize an unsupervised loss function at test-time on the received unlabeled data to improve performance under domain shifts Niu et al. (2022). This includes simple adjustments to the statistics of normalization layers Li et al. (2016), entropy minimization Wang et al. (2020), information maximization Liang et al. (2020), among others Boudiaf et al. (2022); Wang et al. (2022a). However, most TTA methods are proposed to combat covariate domain shifts at test time. In this work, we get inspiration from the source hypothesis adaptation method Liang et al. (2020) to propose an effective regularizer for continual learning. We also analyze the effectiveness of other adaptation methods such as entropy minimization in mitigating catastrophic forgetting in continual learning.

This work aims to enhance continual learning performance by introducing a cost-effective regularizer that improves results even in memory-constrained scenarios. Such scenarios are particularly important when dealing with memory-intensive data, such as videos, or when sample storage is restricted due to privacy concerns. We investigate a class-independent regularizer designed to facilitate the learning of generalizable features.

## 3 Methodology

In this section, we formalize the problem of class-incremental learning in visual recognition tasks. We define the underlying framework and introduce the necessary notation to describe the incremental learning process. Additionally, we present the formulation of the proposed regularizer, Information Maximization (IM), along with the selected baseline regularizers: Elastic Weight Consolidation (EWC), Synaptic Intelligence (SI), and Entropy Minimization (EM).

We focus on the offline continual learning problem for visual recognition tasks, where a classifier $f_\theta : \mathcal{X} \to \mathcal{P}(\mathcal{Y})$ (a DNN parameterized by $\theta$) maps an input $x \in \mathcal{X}$ into the probability simplex[1] $\mathcal{P}(\mathcal{Y})$, with $\mathcal{Y} = \{1, 2, \dots, K\}$. In continual learning, $f_\theta$ is presented with a sequence of $T$ tasks

---

[1] *e.g.* the network's output after Softmax.

$\{(X_1, Y_1), (X_2, Y_2), \ldots, (X_T, Y_T)\}$ where $X_i \subset \mathcal{X}$ and $Y_i \subset \mathcal{Y}$ $\forall i$ Villa et al. (2022). Furthermore, we consider the class-incremental problem setup, where the labels presented in each individual task are mutually exclusive ($Y_i \cap Y_j = \phi \; \forall i \neq j$). The main objective of the learner is to maximize its performance (*e.g.* classification accuracy) on all observed tasks. This objective is often hindered by the catastrophic forgetting problem: while learning task $i$, $f_\theta$ tends to forget previously learned tasks $< i$, significantly dropping its performance for any $x \in X_{<i}$.

For our baseline, we consider rehearsal-based continual learning methods where the learner is allowed to store up to $N$ training examples from previous tasks into a replay memory buffer $M$ Chaudhry et al. (2019b). Let $\mathcal{M}_t$ denote the replay buffer at task $t$ containing examples from the tasks $i < t$. Rehearsal-based methods update the parameter set $\theta$ at task $t$ in the following form:

$$\theta_t^* = \arg\min_\theta \; \mathbb{E}_{(x,y)\sim(X_t, Y_t)}\mathcal{L}(f_\theta(x), y) + \mathbb{E}_{(u,v)\sim M_t}\mathcal{L}(f_\theta(u), v). \tag{1}$$

That is, for each batch sampled from the newly available data on the $t^{th}$ task, the learner samples another batch from memory $\mathcal{M}_t$ and updates the model on the combined loss.

### 3.1 Regularizing Replay Methods with Information Maximization

Inspired by the work of Liang et al. (2020) in the domain of test-time adaptation, we propose that in a continual learning setup, $f_\theta$ should output confident predictions that distinctly separate all previously seen classes. To achieve this, we propose a regularizer that encourages the model to make confident predictions across all encountered classes without biasing its predictions towards recent task data. This means that for any given input, the model should assign a high probability to a single class. We can achieve this by maximizing information in the logits, as a result our approach helps reinforce discriminative representations for all learned classes, improving robustness against distribution shifts. Our proposed regularizer ($\mathcal{R}_{\mathrm{IM}}$) takes the following form:

$$\mathcal{R}_{\mathrm{IM}}(\theta, X_t) = \mathcal{L}_{\mathrm{ent}}(\theta, X_t) + \mathcal{L}_{\mathrm{div}}(\theta, X_t) \tag{2}$$

$$\text{with} \quad \mathcal{L}_{\mathrm{ent}}(\theta, X_t) = -\mathbb{E}_{x\sim X_t} \sum_{k=1}^{K} f_\theta^k(x) \log f_\theta^k(x) \qquad \mathcal{L}_{\mathrm{div}} = \sum_{k=1}^{K} \hat{f}_\theta^k \log \hat{f}_\theta^k,$$

where $\hat{f}_\theta = \mathbb{E}_{x\sim X_t}[f_\theta^k(x)]$ and $f_\theta^k(x)$ is the $k^{th}$ element in the vector $f_\theta(x)$. The IM regularizer is grounded in the principle of mutual information maximization between inputs $X$ and predicted outputs $Y$ Shannon (1948), formalized as $I(X; Y) = H(Y) - H(Y|X)$. This decomposition reveals two complementary objectives: (1) minimizing conditional entropy $H(Y|X)$ ensures confident predictions for each input (achieved by $\mathcal{L}_{ent}$), and (2) maximizing marginal entropy $H(Y)$ ensures predictions are distributed across all output classes (achieved by $\mathcal{L}_{div}$). Without $\mathcal{L}_{div}$, the model could trivially minimize $\mathcal{L}_{ent}$ by assigning high-confidence predictions to a single class for all inputs, causing prediction collapse. $\mathcal{L}_{div}$ prevents this by forcing the model's average predictions to span the entire label space [2]. Together, these terms encourage confident yet balanced predictions across all classes. Our regularized rehearsal-based method integrates this IM regularizer into the continual learning objective as follows:

$$\min_\theta \; \mathbb{E}_{(x,y)\sim(X_t, Y_t)}\mathcal{L}(f_\theta(x), y) + \mathbb{E}_{(u,v)\sim M_t}\mathcal{L}(f_\theta(u), v) + \mathcal{R}_{\mathrm{IM}}(\theta, X_t). \tag{3}$$

Our proposed regularizer has the following advantages: **(i)** It is orthogonal to the most critical design choices of continual learning algorithms, as it can operate regardless of the choice of $f_\theta$, the replay-based method, the size of the memory buffer, and the number of tasks. **(ii)** Efficient computation of $\mathcal{R}_{IM}$: where both $\mathcal{L}_{\mathrm{ent}}$ and $\mathcal{L}_{\mathrm{div}}$ depend exclusively on the output predictions of the model and can be computed in $\mathcal{O}(n)$. This aspect is essential when dealing with memory-intensive setups. For example, on video data, our regularizer

---

[2]The subscripts "ent" stands for entropy and "div" stands for diversity

estimates $\mathcal{L}_{\mathrm{ent}}$ and $\mathcal{L}_{\mathrm{div}}$ over clip predictions instead of per-frame estimates. **(iii)** Our formulation is agnostic to the type of data used in the continual learning problem. Without any modifications, our formulation can be applied to both image-based or video-based continual learning problems.

### 3.2 Baseline Regularizers

We compare our proposal against different regularizers to assess its effectiveness in mitigating forgetting and improving continual learning performance. We follow the formulation in Equation (3), and study alternatives to $\mathcal{R}_{\mathrm{IM}}(\theta, X_t)$. In particular, we analyze different regularizers from the continual learning literature, namely Elastic Weight Consolidation Kirkpatrick et al. (2017) and Synaptic Intelligence Zenke et al. (2017). Furthermore, we explore Entropy Minimization Wang et al. (2020) from the test-time adaptation literature.

**Elastic Weight Consolidation (EWC).** Kirkpatrick et al. (2017) proposed to regularize the parameter update during continual learning to prevent catastrophic forgetting by constraining changes to important weights. The key idea behind EWC is to estimate the importance of each parameter for previously learned tasks and penalize deviations from their learned values. We analyze the effectiveness of combining EWC with rehearsal-based methods by replacing $\mathcal{R}_{\mathrm{IM}}$ in Equation (3) with $\mathcal{R}_{\mathrm{EWC}}$, defined as:

$$\mathcal{R}_{\mathrm{EWC}}(\theta) = \sum_i \frac{\lambda}{2} F_i (\theta^i - \theta^i_{t-1})^2,$$

where $F_i$ denotes the $i$-th diagonal element of the Fisher information matrix $F$, which quantifies the importance of the $i$-th parameter based on how sensitive the loss function is to changes in that parameter. Here, $\theta^i$ and $\theta^i_{t-1}$ are the $i$-th parameters at the current task (we are optimizing for) and previous task, respectively, and $\lambda$ is a hyperparameter balancing the relative importance of old tasks with respect to the current task.

**Synaptic Intelligence (SI).** It is a biologically inspired regularizer from the continual learning literature. It follows a similar principle to EWC but determines weight importance using a different approach. Instead of using the Fisher Information Matrix, SI tracks the contribution of each parameter during training by accumulating an importance measure based on changes in loss. This adaptive tracking mechanism allows the model to selectively constrain updates to crucial parameters while remaining flexible for learning new tasks. We replace $\mathcal{R}_{\mathrm{IM}}$ in Equation (3) with $\mathcal{R}_{\mathrm{SI}}$ which takes the following form:

$$\mathcal{R}_{\mathrm{SI}}(\theta) = \sum_{t=1}^{T} \frac{\omega_t^k}{(\Delta\theta_k^t)^2 + \xi},$$

where $\Delta\theta_k^t = \theta_k^t - \theta_k^{t-1}$ is the change in the $k$-th parameter between task $t$ and task $t-1$, $\omega_t^k$ is the per-parameter importance weight for parameter $k$ at task $t$, $k$ indexes the parameters, $t$ indexes the tasks, and the damping parameter $\xi$ avoids division by zero.

**Entropy Minimization (EM).** Following the self-supervised spirit of our proposed regularization approach, we include one self-supervised regularizer that encourages the model to produce more confident predictions. In particular, we follow Wang et al. (2020) and apply entropy minimization to regularize the output distribution, reducing the model's uncertainty when making predictions. Entropy minimization replaces $\mathcal{R}_{\mathrm{IM}}$ in Equation (3) with $\mathcal{R}_{\mathrm{EM}}$ where:

$$\mathcal{R}_{\mathrm{EM}}(\theta, X_t) = -\mathbb{E}_{x \sim X_t} \sum_{k=1}^{K} f_\theta^k(x) \log f_\theta^k(x) = \mathcal{L}_{\mathrm{ent}}(\theta, X_t). \tag{4}$$

Entropy minimization encourages the model to assign higher confidence to its predictions, effectively suppressing uncertain outputs. This can be beneficial in a continual learning setup, where distribution shifts can lead to increased uncertainty. This entropy minimization term forms one component of our proposed IM regularizer ($\mathcal{L}_{\mathrm{ent}}$).

## 4  Experiments

In this section, we proceed with the empirical assessment of our proposed approach to validate its effectiveness. For completeness, we first evaluate several rehearsal-based continual learning (CL) methods (ER, DER, and DER++) when paired with the regularizers IM, EWC, SI, and EM. We then extend the analysis by applying IM to more advanced rehearsal approaches (Refresh Learning and STAR), showing that its benefits generalize consistently across different methods and datasets.

**Datasets.**   Following the image CL literature, we focus on two main datasets: Split-CIFAR100 Zenke et al. (2017) and Split-Tiny ImageNet Le & Yang (2015). Split-CIFAR100 contains a total of 100 classes and 6000 images per class. It is divided into 10 tasks, each containing 10 classes. Split-Tiny ImageNet consists of 200 classes with 500 images per class, and is divided into 10 tasks of 20 classes each.

**Evaluation Metrics.**   To evaluate the performance of CL methods, we consider two metrics: Average Accuracy, which is defined as the average performance across all tasks, and Forgetting Rate that measures the impact of the learned task on the performance of the previous tasks Chaudhry et al. (2019a). These metrics provide complementary perspectives on the effectiveness of each approach in balancing stability and adaptability in a continual learning setting.

- **Average Accuracy (ACC)** quantifies the model's overall performance across all tasks it has encountered. It is defined as:

$$ACC = \frac{1}{T} \sum_{i=1}^{T} a_i, \tag{5}$$

  where $T$ is the total number of tasks and $a_i$ represents the accuracy of the model on the $i$-th task after it has been trained on all $T$ tasks. ACC provides a comprehensive measure of how well the model learns and retains knowledge across a full sequence of tasks.

- **Forgetting Rate (FR)** measures the decrease in performance on past tasks after a model has been trained on new ones. It directly measures catastrophic forgetting. FR is defined as:

$$FR = \frac{1}{T-1} \sum_{i=1}^{T-1} \max_{j<T} (a_{ij} - a_{iT}), \tag{6}$$

  where $a_{ij}$ is the accuracy on task $i$ immediately after training on task $j$, and $a_{iT}$ is the accuracy on task $i$ after the final task $T$ has been learned. A lower FR indicates better retention of previously learned knowledge, while a higher FR points to significant forgetting.

**Implementation Details.**   We train a ResNet18 He et al. (2016) model from scratch and summarize the training parameters in the **Appendix**. Following our limited memory setting, we define a budget of 5, 10 and 20 samples per class as the maximum allowed in $\mathcal{M}_t$ at any moment. We integrate the different regularizers into the Mammoth CL framework Boschini et al. (2022); Buzzega et al. (2020a). To balance the loss terms, we multiply the regularization term by $\lambda$=0.5 and the cross-entropy loss with 1-$\lambda$. All main experiments employ a fixed random seed to ensure fair comparison across methods. Further details on hyperparameter sensitivity, training parameters, and statistical validation are provided in the **Appendix**.

**Baselines.**   We consider four regularizers: EWC Kirkpatrick et al. (2017), SI Zenke et al. (2017), EM Wang et al. (2020), and IM Liang et al. (2020), applied on top of three memory-based continual learning methods: ER Rolnick et al. (2019), DER Buzzega et al. (2020b), and DER++ Buzzega et al. (2020b). For the IM regularizer, we further extend the evaluation to include Refresh Learning Wang et al. (2024), implemented on top of DER++, and STAR Eskandar et al. (2025), implemented on top of ER.

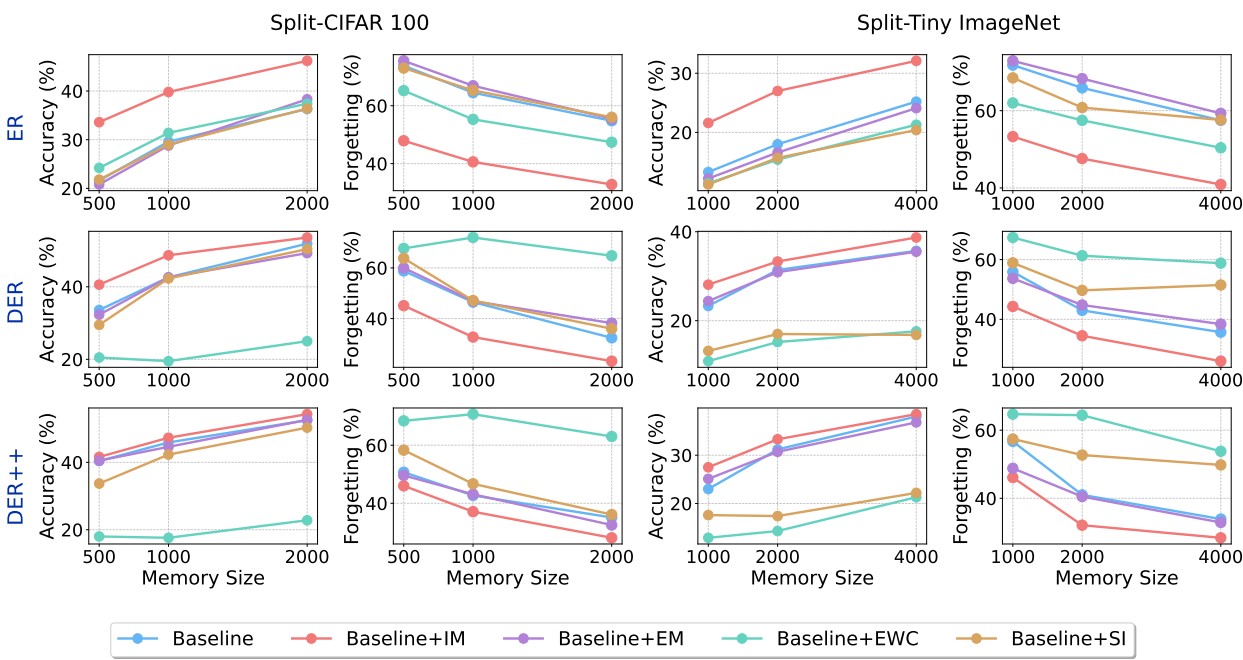

Figure 1: **Results of Integrating Different Regularizers on Split-CIFAR100 and Split-Tiny ImageNet.** This figure plots the average accuracy and forgetting rate of three baseline methods (ER, DER, and DER++) across various sizes of memory buffer, in combination with the analyzed regularizers (IM, EM, EW, and SI), and across two datasets (Split-CIFAR100 and Split-Tiny ImageNet). The results demonstrate that the proposed information maximization regularizer (IM) consistently outperforms other methods, achieving higher accuracy and lower forgetting rates on both datasets regardless of the memory setting.

## 4.1 Regularized Rehearsal Methods Results

Figure (1) summarizes the performance of rehearsal-based methods ER, DER, and DER++ on Split-CIFAR100 (left columns) and Split-Tiny ImageNet (right columns) for the selected memory sizes. We include the baseline performance (light blue) and outline the impact of incorporating regularization techniques on top of these rehearsal methods.

Our results demonstrate that introducing IM on top of rehearsal-based methods consistently leads to improvements across all memory sizes. For instance, when IM is applied to ER on Split-CIFAR100, we observe an enhancement of 10-13% in performance across all memory sizes. The improvement is slightly lower for DER and DER++, ranging around 2-7% for DER and 1-2% for DER++. In contrast, other regularizers like EWC, SI, and EM generally do not improve the baseline methods, and can even degrade performance in some cases, as is the case for EWC on DER and DER++, where the model's accuracy drops by nearly half.

We can also observe in Figure (1) that forgetting is significantly reduced when the rehearsal methods are paired with IM. For ER (paired with IM) applied on Split-CIFAR100, the reduction in forgetting is around (22-25%) across all memory sizes. For DER and DER++ on Split-CIFAR100, our results show a smaller reduction compared to ER with about (9-13%) and (4-6%), respectively. On the other hand, EM and SI do not generally reduce forgetting compared to the original baselines. This is since EM aims at increasing the model's confidence in predicting samples from the current task. While this approach might accelerate the learning process over tasks, it does not promote the retention of previously learned information.

For EWC, we observe reduction in forgetting when paired with ER, but not with the remaining baselines. More detailed and comprehensive results can be found in the **Appendix**.

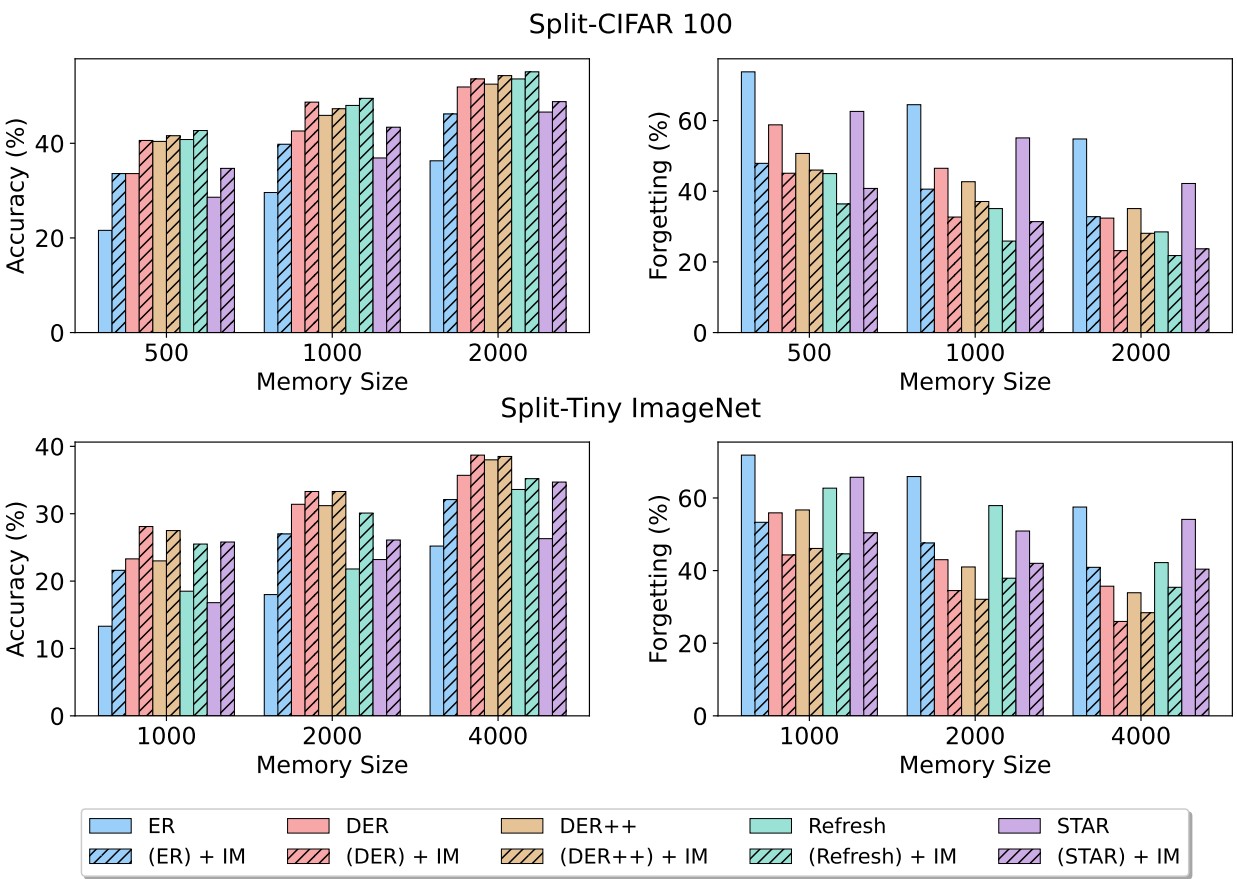

Figure 2: **Results on Split-CIFAR100 and Split-Tiny ImageNet.** This figure presents the average accuracy and forgetting rate of five rehearsal-based methods (ER, DER, DER++, Refresh Learning, and STAR) across different memory buffer sizes, both in their baseline form and when combined with Information Maximization (IM). The results show that integrating IM consistently enhances performance, leading to higher accuracy and reduced forgetting across all methods, datasets, and memory settings.

To further validate the effectiveness of IM, we expand our analysis to include two additional rehearsal-based baselines: Refresh Learning and STAR. Figure (2) presents the performance gains when IM is integrated into all five rehearsal methods (ER, DER, DER++, Refresh Learning, STAR) across both Split-CIFAR100 and Split-Tiny ImageNet.

On Split-CIFAR100, IM consistently improves the performance of Refresh Learning and STAR, but with varying impact. When paired with Refresh Learning, the improvements are more modest, about 1–2% in accuracy, but forgetting is reduced by (7–9%) across all memory sizes. This indicates that while Refresh Learning already stabilizes training to some degree, IM provides an additional layer of retention without significantly altering the learning dynamics. For STAR, the effect is more significant, as we observe accuracy gains of 2–6%, along with a substantial reduction in forgetting (18–24%).

On Split-Tiny ImageNet, IM continues to yield consistent improvements. For Refresh Learning, the accuracy gains are larger than on Split-CIFAR100, ranging from 2–8%, with forgetting reduced by (7–20%). STAR also benefits considerably, with accuracy improvements of 3–9% and forgetting reductions between (9–15%).

**Conclusion.** These results reveal that Information Maximization (IM) serves as an effective regularization technique, consistently improving the performance of image continual learning baselines across various mem-

Table 1: **Ablation Study on Compute Budget.** This table presents the performance of ER and DER, with and without the Information Maximization (IM) regularizer, on Split-CIFAR100 and Split-Tiny ImageNet datasets. For Split-CIFAR100, the baselines are run with 10 epochs instead of the original 50, while for Split-Tiny ImageNet, the experiments are run with 50 epochs instead of the original 100. The results indicate that incorporating IM leads to improved convergence and consistently higher average accuracy.

| | Split-Cifar100 | | | Split-Tiny ImageNet | | |
|---|---|---|---|---|---|---|
| **Buffer Size** | **500** | **1000** | **2000** | **1000** | **2000** | **4000** |
| ER | 20.8 | 26.8 | 35.6 | 13.2 | 18.7 | 25.6 |
| ER + IM | **28.8** | **35.7** | **40.1** | **21.5** | **26.6** | **31.8** |
| DER | 24.1 | 21.1 | 19.9 | 21.6 | 26.3 | 24.2 |
| DER + IM | **33.5** | **37.3** | **34.4** | **27.9** | **32.1** | **33.3** |

ory budgets. By encouraging confident yet balanced predictions, IM enhances both accuracy and knowledge retention, thereby mitigating catastrophic forgetting.

## 4.2 Ablation Analysis

To explore the limitations of Information Maximization (IM) as a regularizer for continual learning methods, we conduct three ablation experiments aimed at understanding its performance under various conditions. First, we assess the impact of IM when the computational budget is reduced to determine whether it improves the convergence of the baseline methods. Second, we evaluate if the improvement obtained by using IM diminishes with additional tasks. Specifically, we investigate the performance of our approach when the number of tasks exceeds 10, to understand how IM scales with an increasing number of tasks. Third, we examine the effect of applying IM exclusively to buffer samples, as opposed to both buffer and current task samples, to assess whether applying IM beyond current task samples is beneficial.

**Computational Budget.** In the experiments presented in Section (4.1), the computational budget was set following the default settings of Mammoth CL framework, *i.e.* 50 epochs per task for Split-CIFAR100 and 100 epochs per task for Split-Tiny ImageNet. These settings ensure sufficient training iterations for each task, allowing models to stabilize and learn meaningful representations. However, in practical scenarios, computational efficiency is a critical factor, as training deep learning models over multiple tasks can be prohibitively expensive. Currently, there is a growing interest in exploring low-computational regimes Ghunaim et al. (2023); Prabhu et al. (2023) for CL, as computing resources are far more expensive than storage Prabhu et al. (2023). This shift in focus reflects the necessity of developing continual learning approaches that remain effective even with reduced training time. Therefore, to investigate how our proposed regularizer, IM, performs under computational constraints, we run ablation experiments with about half of the computational budget on Split-Tiny ImageNet (*i.e.* half of the training epochs per task) and less than a quarter of the computational budget on Split-CIFAR100 compared to our original experimental setup.

The results in Table (1) show that, even with a lower computational budget of 10 and 50 epochs per task on Split-CIFAR100 and Split-Tiny ImageNet, respectively, the proposed ER+IM and DER+IM methods outperform their counterparts without IM regularizer. For instance, on the Split-CIFAR100 dataset with a buffer size of 1000, ER+IM achieves an accuracy of 35.7%, significantly higher than ER at 26.8%. Similarly, DER+IM attains 37.3% accuracy, surpassing DER's 21.1% by a large margin. These trends hold for different buffer sizes and datasets, highlighting the effectiveness of the proposed regularization in low-compute regimes.

**Number of Tasks.** In the experiments presented in Section (4.1), we used the conventional 10-tasks split for Split-CIFAR100 and Split-Tiny ImageNet, which is commonly used in continual learning studies. However, as shown in Villa et al. (2022); Prabhu et al. (2023), performance may vary when more tasks are introduced. More tasks can make the problem harder because the model has to remember more information

Table 2: **Ablation Study on Number of Tasks.** This table presents the performance of ER and DER , with and without Information Maximization (IM) regularizer, on a sequence of 20 tasks for both Split-CIFAR100 and Split-Tiny ImageNet datasets. Our evaluation shows that, despite the increased amount of tasks our proposal still outperform the ER and DER baselines in every single scenario.

| | Split-CIFAR100 | | | Split-Tiny ImageNet | | |
| --- | --- | --- | --- | --- | --- | --- |
| **Buffer Size** | **500** | **1000** | **2000** | **1000** | **2000** | **4000** |
| ER | 16.6 | 25.8 | 34.7 | 9.1 | 14.5 | 22.1 |
| ER + IM | **23.4** | **31.3** | **38.9** | **18.3** | **23.4** | **29.1** |
| DER | 25.1 | 35.8 | 38.9 | 18.0 | 22.5 | 27.8 |
| DER + IM | **32.8** | **39.8** | **45.0** | **23.2** | **27.0** | **32.1** |

and avoid forgetting earlier tasks while learning new information. Consequently, we reran the experiments in Section (4.1), doubling the number of tasks from 10 to 20. This allows us to evaluate whether Information Maximization (IM) regularizer remains effective when the continual learning problem becomes more challenging due to having more tasks to learn. The results presented in Table (2) show that incorporating IM into ER and DER can significantly improve their performance on longer sequences of tasks. For example, ER+IM shows an improvement of (4-7%) and (6-8%) on Split-CIFAR100 and Split-Tiny ImageNet, respectively. On the other hand, DER+IM shows a (4-7%) and (4-5%) improvement on Split-CIFAR100 and Split-Tiny ImageNet, respectively.

**Regularization Targets.** For the experimental assessment in Section (4.1), we apply the regularization loss to current task samples only. This raises the question of how the proposed method would behave if the IM loss were applied exclusively to memory samples or to both memory and current task samples. For this reason, we reran the experiments shown in Section (4.1) for both variants, and the results are summarized in Table (3). We find that applying the IM loss to the current task (CT) is superior to applying it to the memory/buffer samples only (BF) or to both buffer and current task samples (ALL). Notably, this trend remains consistent across various buffer sizes, datasets, and continual learning methods, highlighting the robustness of this strategy.

For example, with a buffer size of 500 on the Split-CIFAR100, ER+IM (CT) achieves an accuracy of 33.6%, which is significantly higher than ER+IM (ALL) and ER+IM (BF), which achieve 25.9% and 21.7%, respectively. Similarly, DER+IM (CT) consistently outperforms DER+IM (ALL) and DER+IM (BF) across various buffer sizes and datasets, reinforcing the advantage of applying IM solely to current task samples. For example, it achieves 48.7% accuracy on Split-CIFAR100 with a buffer size of 1000, while DER +IM (ALL) and DER +IM (BF) achieve 46.0% and 41.0%, respectively. These results indicate that applying IM loss to the current task samples is more effective than applying it exclusively to the memory/buffer samples only or to both buffer and current task samples (refer to **Appendix** for further analysis).

## 4.3 Generalization to Video Continual Learning

To further validate the effectiveness of Information Maximization (IM) as a cost-effective regularization technique for enhancing continual learning methods, we extend our analysis to representative experiments in the video domain. Specifically, we experiment with the iCaRL approach as part of the popular vCLIMB framework Rebuffi et al. (2017); Villa et al. (2022) to assess IM's performance in this video CL context.

We evaluate the use of IM loss on two widely recognized datasets in the video domain: UCF-101, consisting of 101 classes split into 10 tasks, and ActivityNet, comprising 200 classes also divided into 10 tasks. We set the memory size to 5, 10 and 20 samples per class and adopt the training hyperparameters from Villa et al. (2022).

Table 3: **Ablation Study on Regularization Target Selection.** In the main results, the Information Maximization (IM) regularizer is applied exclusively to the current task samples (CT). This table presents the results of applying the regularizer to the buffer samples only (BF) and to both current task samples and buffer samples simultaneously (ALL). The findings indicate that applying the regularizer to the current task samples consistently leads to superior performance compared to the other variants.

| Buffer Size | Split-CIFAR100 | | | Split-Tiny ImageNet | | |
|---|---|---|---|---|---|---|
| | 500 | 1000 | 2000 | 1000 | 2000 | 4000 |
| ER + IM (ALL) | 25.9 | 34.6 | 42.7 | 15.0 | 20.3 | 27.8 |
| ER + IM (CT) | **33.6** | **39.8** | **46.2** | **21.6** | **27.0** | **32.1** |
| ER + IM (BF) | 21.7 | 28.9 | 38.8 | 12.7 | 17.7 | 24.2 |
| DER + IM (ALL) | 33.5 | 46.0 | 53.5 | 27.7 | 32.7 | 35.1 |
| DER + IM (CT) | **40.6** | **48.7** | **53.6** | **28.1** | **33.3** | **38.7** |
| DER + IM (BF) | 27.3 | 41.0 | 50.2 | 21.2 | 27.7 | 34.6 |

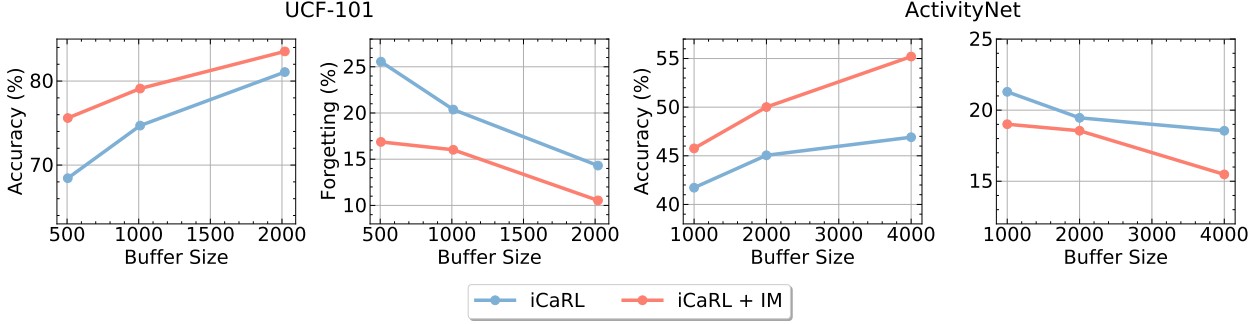

Figure 3: **Application of Information Maximization to Video Continual Learning.** This figure illustrates the average accuracy and forgetting rates of the iCARL video continual learning variant, introduced by vCLIMB Villa et al. (2022), with and without our Information Maximization (IM) regularizer. The results demonstrate that incorporating the IM regularizer on top of iCARL leads to consistent improvements in average accuracy and reductions in the forgetting rate.

Upon applying iCARL to UCF-101 with IM, we achieve an improvement of 2-8% across all memory sizes. Similarly, on ActivityNet, the accuracy gain is between 4-8% with the incorporation of IM. These results further underscore the potential of IM as a valuable regularization technique for enhancing performance in the video domain and continual learning scenarios. Note that video data has an additional temporal dimension compared to images, which requires more memory to store. Being able to improve continual learning performance with small memory buffer sizes, as shown in Figure 3, is crucial for facilitating the development of memory-efficient approaches for video continual learning.

## 5 Conclusion

In conclusion, this paper explores the combined potential of memory-based methods and regularization techniques in the context of Continual Learning (CL), specifically within a class incremental setup. We introduce a novel, class-agnostic regularization strategy for CL, which focuses on the distribution of the network's predictions. This strategy, termed Information Maximization (IM) regularization, facilitates the

learning of enhanced feature representations across multiple distribution shifts, while simultaneously minimizing memory requirements and computational overhead. Our extensive empirical evaluation underscores the effectiveness of the proposed IM regularizer, demonstrating its superiority over existing regularization strategies designed for CL. Furthermore, the simplicity and versatility of our approach allow it to be applied across different input domains, as evidenced by its successful application in the video continual learning setup. Unlike traditional image-based settings, video CL presents additional challenges due to its temporal structure and higher memory demands. Despite these complexities, our method demonstrates strong performance, reinforcing its applicability to real-world, resource-constrained scenarios.

## Acknowledgment

This work is supported by the KAUST Center of Excellence for Generative AI under award number 5940. The computational resources are provided by IBEX, which is managed by the Supercomputing Core Laboratory at KAUST.

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

## 6 Appendix

### 6.1 Regularized Rehearsal Methods Results

Tables 4 and 5, contain the numerical results for the average accuracy and forgetting rate metrics, respectively, on three baseline methods (ER, DER, and DER++) in combination with the analyzed regularizers (IM, EM, EW, and SI), as well as for integrating IM into Refresh Learning and STAR. These results were summarized as plots in Figures (1) and (2) of the main paper. We observe that our proposed regularize (IM) consistently outperforms other methods across different memory settings. The hyper-parameters we used for every baseline when combined with the analyzed regularizers (IM, EM, EW, and SI) are listed in Table 6 and 7.

Table 4: **Average Accuracy on Split-CIFAR100 and Split-Tiny ImageNet.** This table shows the average accuracy of three baseline methods (ER, DER, and DER++) across various sizes of memory buffer, in combination with the analyzed regularizers (IM, EM, EW, and SI), as well as for integrating IM into Refresh Learning and STAR.

| Dataset | Split-CIFAR100 | | | Split-Tiny ImageNet | | |
|---|---|---|---|---|---|---|
| Buffer | 500 | 1000 | 2000 | 1000 | 2000 | 4000 |
| ER | 21.6 | 29.6 | 36.3 | 13.3 | 18.0 | 25.2 |
| ER (IM) | **33.6** | **39.8** | **46.2** | **21.6** | **27.0** | **32.1** |
| ER (EM) | 20.8 | 28.8 | 38.3 | 12.2 | 16.6 | 24.1 |
| ER (EWC) | 24.2 | 31.4 | 37.4 | 11.4 | 15.4 | 21.3 |
| ER (SI) | 21.8 | 29.0 | 36.4 | 11.2 | 15.7 | 20.4 |
| DER | 33.6 | 42.6 | 51.9 | 23.3 | 31.4 | 35.7 |
| DER (IM) | **40.6** | **48.7** | **53.6** | **28.1** | **33.3** | **38.7** |
| DER (EM) | 32.3 | 42.6 | 49.3 | 24.4 | 30.9 | 35.5 |
| DER (EWC) | 20.5 | 19.5 | 25.0 | 10.9 | 15.2 | 17.6 |
| DER (SI) | 29.5 | 42.3 | 50.4 | 13.2 | 17.0 | 16.8 |
| DER++ | 40.4 | 45.9 | 52.5 | 23.0 | 31.2 | 38.0 |
| DER++ (IM) | **41.6** | **47.3** | **54.3** | **27.5** | **33.3** | **38.5** |
| DER++ (EM) | 40.5 | 44.6 | 52.6 | 25.1 | 30.7 | 36.8 |
| DER++ (EWC) | 18.0 | 17.6 | 22.8 | 12.9 | 14.3 | 21.3 |
| DER++ (SI) | 33.7 | 42.3 | 50.3 | 17.6 | 17.4 | 22.2 |
| Refresh | 40.8 | 48.0 | 53.6 | 18.5 | 21.8 | 33.6 |
| Refresh (IM) | **42.7** | **49.5** | **55.1** | **25.5** | **30.1** | **35.2** |
| STAR | 28.6 | 36.9 | 46.6 | 16.8 | 23.2 | 26.3 |
| STAR (IM) | **34.7** | **43.4** | **48.8** | **25.8** | **26.1** | **34.7** |

### 6.2 Generalization to Video Continual Learning

Tables 8 and 9 present the numerical results of average accuracy and forgetting rate, respectively, for iCaRL approach with and without IM on two video datasets (UCF101 and ActivityNet). Combining iCaRL with IM shows improvement in average accuracy and reduces forgetting rate across different memory settings.

Table 5:   **Forgetting Rate on Split-CIFAR100 and Split-Tiny ImageNet.** This table shows the forgetting rate of three baseline methods (ER, DER, and DER++) across various sizes of memory buffer, in combination with the analyzed regularizers (IM, EM, EW, and SI), as well as the results for integrating IM into Refresh Learning and STAR.

| Dataset | Split-CIFAR100 | | | Split-Tiny ImageNet | | |
|---|---|---|---|---|---|---|
| Buffer | 500 | 1000 | 2000 | 1000 | 2000 | 4000 |
| ER | 73.8 | 64.5 | 54.8 | 71.8 | 65.9 | 57.5 |
| ER (IM) | **47.9** | **40.6** | **32.8** | **53.3** | **47.6** | **40.9** |
| ER (EM) | 75.5 | 66.9 | 55.5 | 72.9 | 68.3 | 59.3 |
| ER (EWC) | 65.2 | 55.3 | 47.4 | 62.0 | 57.5 | 50.4 |
| ER (SI) | 73.0 | 65.3 | 56.0 | 68.5 | 60.8 | 57.6 |
| DER | 58.8 | 46.5 | 32.4 | 55.9 | 43.0 | 35.7 |
| DER (IM) | **45.1** | **32.7** | **23.2** | **44.3** | **34.5** | **26.0** |
| DER (EM) | 60.0 | 46.9 | 38.2 | 53.7 | 44.8 | 38.4 |
| DER (EWC) | 67.7 | 72.0 | 64.8 | 67.4 | 61.3 | 58.8 |
| DER (SI) | 63.8 | 47.1 | 36.0 | 58.9 | 49.7 | 51.5 |
| DER++ | 50.7 | 42.7 | 35.1 | 56.7 | 41.0 | 33.9 |
| DER++ (IM) | **46.0** | **37.1** | **28.1** | **46.1** | **32.1** | **28.4** |
| DER++ (EM) | 49.6 | 43.1 | 32.5 | 48.8 | 40.5 | 32.9 |
| DER++ (EWC) | 68.4 | 70.7 | 63.0 | 64.7 | 64.4 | 53.8 |
| DER++ (SI) | 58.3 | 46.7 | 36.1 | 57.4 | 52.7 | 49.8 |
| Refresh | 45.0 | 35.1 | 28.5 | 62.7 | 57.9 | 42.2 |
| Refresh (IM) | **36.4** | **25.9** | **21.8** | **44.6** | **37.9** | **35.4** |
| STAR | 62.6 | 55.1 | 42.2 | 65.7 | 50.9 | 54.1 |
| STAR (IM) | **40.8** | **31.4** | **23.7** | **50.4** | **42.0** | **40.4** |

### 6.3   Sensitivity Analysis: Regularization Strength ($\lambda$)

We provide a detailed sensitivity analysis of the regularization strength ($\lambda$) to demonstrate the robustness of our findings and address concerns regarding fair hyperparameter comparison. We show that IM consistently outperforms baseline regularizers across a range of settings, confirming that our main conclusions are not artifacts of a specific hyperparameter choice. For the primary results presented in the paper, we adopted a fixed hyperparameter protocol, using a consistent regularization strength ($\lambda = 0.5$) for all regularizers (IM, EM, EWC, and SI). This protocol ensures a direct and fair test of robustness by evaluating all methods under identical experimental conditions. To validate that our findings are not artifacts of this fixed $\lambda$ choice, we conducted a sensitivity analysis across $\lambda \in 0.2, 0.5, 0.8$ on Split-CIFAR100 with a memory buffer of 500 samples. The results for Experience Replay (ER) and Dark Experience Replay (DER) are presented in Table 10 and Table 11, respectively. The analysis demonstrates that our proposed IM regularizer substantially outperforms all other regularizers, even when each method uses its individually optimal $\lambda$ value. These results confirm that our conclusions are robust to hyperparameter choices and are not an artifact of favoring IM through biased selection.

### 6.4   Regularization Target Selection

A critical design choice in the application of the Information Maximization (IM) regularizer concerns the selection of the target samples to which the regularization is applied. Three natural candidates emerge: (i) Current Task (CT) samples only, (ii) Buffer (BF) samples only, and (iii) All (ALL) samples, encompassing both current task and buffer samples. While intuition might suggest that regularizing the rehearsal buffer

Table 6: **Experiments Hyper-Parameters (Split-CIFAR100).** Columns list the training hyper-parameters. Buf. = buffer size, LR = learning rate, Mom. = momentum, WD = weight decay, $\alpha$ = distillation coefficient (used in DER/DER++/Refresh), $\beta$ = memory balancing coefficient (used in DER++/Refresh), $\gamma$, $\lambda$, $s$ (specific to STAR), $\Gamma$, $J$ (specific to Refresh Learning). A dash ($-$) indicates that the parameter is not applicable to the method.

| Baseline | Buf. | LR | Mom. | WD | $\alpha$ | $\beta$ | $\gamma$ | $\lambda$ | $s$ | $\Gamma$ | $J$ |
|---|---|---|---|---|---|---|---|---|---|---|---|
| ER | 500 | 0.1 | 0 | 0 | $-$ | $-$ | $-$ | $-$ | $-$ | $-$ | $-$ |
| ER | 1000 | 0.1 | 0 | 0 | $-$ | $-$ | $-$ | $-$ | $-$ | $-$ | $-$ |
| ER | 2000 | 0.1 | 0 | 0 | $-$ | $-$ | $-$ | $-$ | $-$ | $-$ | $-$ |
| DER | 500 | 0.03 | 0 | 0 | 0.3 | $-$ | $-$ | $-$ | $-$ | $-$ | $-$ |
| DER | 1000 | 0.03 | 0 | 0 | 0.3 | $-$ | $-$ | $-$ | $-$ | $-$ | $-$ |
| DER | 2000 | 0.03 | 0 | 0 | 0.3 | $-$ | $-$ | $-$ | $-$ | $-$ | $-$ |
| DER++ | 500 | 0.03 | 0 | 0 | 0.3 | 0.5 | $-$ | $-$ | $-$ | $-$ | $-$ |
| DER++ | 1000 | 0.03 | 0 | 0 | 0.3 | 0.5 | $-$ | $-$ | $-$ | $-$ | $-$ |
| DER++ | 2000 | 0.03 | 0 | 0 | 0.3 | 0.5 | $-$ | $-$ | $-$ | $-$ | $-$ |
| Refresh | 500 | 0.03 | 0 | 1e-4 | 0.3 | 0.5 | $-$ | $-$ | $-$ | 1e-5 | 1 |
| Refresh | 1000 | 0.03 | 0 | 1e-4 | 0.3 | 0.5 | $-$ | $-$ | $-$ | 1e-5 | 1 |
| Refresh | 2000 | 0.03 | 0 | 1e-4 | 0.3 | 0.5 | $-$ | $-$ | $-$ | 1e-5 | 1 |
| STAR | 500 | 0.1 | 0 | 0 | $-$ | $-$ | 0.05 | 0.05 | 1 | $-$ | $-$ |
| STAR | 1000 | 0.05 | 0 | 0 | $-$ | $-$ | 0.05 | 0.05 | 1 | $-$ | $-$ |
| STAR | 2000 | 0.05 | 0 | 0 | $-$ | $-$ | 0.05 | 0.05 | 1 | $-$ | $-$ |

should be most effective for mitigating catastrophic forgetting, our empirical findings reveal a counter-intuitive result: applying the IM regularizer exclusively to current task samples yields substantially superior performance (Table 3).

**Feature Drift Analysis.** To understand why CT-only regularization outperforms BF-only regularization, we conducted a feature drift analysis. Feature drift is quantified as the mean L2 distance between class-specific feature representations (penultimate layer activations) immediately after learning a task versus after all tasks are completed. We report the feature drift for the first two tasks (Task 1 and Task 2).

The feature drift analysis in Table 12 reveals a striking and counterintuitive finding: both IM-CT and IM-BF increase feature drift compared to the baseline, yet only IM-CT yields improved performance. This observation shows that effective continual learning regularization does not necessarily minimize feature drift; instead, it can enable controlled feature evolution that maintains class discriminability.

The per-task breakdown in Table 13 shows that IM-CT consistently improves old task accuracy (T1-T9), while sacrificing recent task performance (T10: 92.0% → 77.2%). This stability-plasticity trade-off is characteristic of effective continual learning. In contrast, IM-BF maintains high performance on the recent task (T10: 93.2%) but fails to prevent forgetting on old tasks, as effectively as IM-CT, indicating that the consistent improvement pattern of IM-CT reflects systematic feature refinement rather than random effects.

## 6.5 Experimental Methodology and Statistical Validation

Our experimental methodology involved two complementary approaches to balance rigor with computational feasibility. First, we conducted preliminary experiments with multiple random seeds to validate observed behaviors and identify consistent trends across different initializations. Second, given the extensive scope of

Table 7: **Experiments Hyper-Parameters (Split-Tiny ImageNet).** Columns list the training hyper-parameters. Buf. = buffer size, LR = learning rate, Mom. = momentum, WD = weight decay, $\alpha$ = distillation coefficient (used in DER/DER++/Refresh), $\beta$ = memory balancing coefficient (used in DER++/Refresh), $\gamma$, $\lambda$, $s$ (specific to STAR), $\Gamma$, $J$ (specific to Refresh Learning). A dash ($-$) indicates that the parameter is not applicable to the method.

| Baseline | Buf. | LR | Mom. | WD | $\alpha$ | $\beta$ | $\gamma$ | $\lambda$ | $s$ | $\Gamma$ | $J$ |
|---|---|---|---|---|---|---|---|---|---|---|---|
| ER | 500 | 0.03 | 0 | 0 | – | – | – | – | – | – | – |
| ER | 1000 | 0.03 | 0 | 0 | – | – | – | – | – | – | – |
| ER | 2000 | 0.03 | 0 | 0 | – | – | – | – | – | – | – |
| DER | 500 | 0.03 | 0 | 0 | 0.1 | – | – | – | – | – | – |
| DER | 1000 | 0.03 | 0 | 0 | 0.1 | – | – | – | – | – | – |
| DER | 2000 | 0.03 | 0 | 0 | 0.1 | – | – | – | – | – | – |
| DER++ | 500 | 0.03 | 0 | 0 | 0.3 | 0.5 | – | – | – | – | – |
| DER++ | 1000 | 0.03 | 0 | 0 | 0.3 | 0.5 | – | – | – | – | – |
| DER++ | 2000 | 0.03 | 0 | 0 | 0.3 | 0.5 | – | – | – | – | – |
| Refresh | 500 | 0.03 | 0 | 1e-4 | 0.3 | 0.5 | – | – | – | 1e-5 | 1 |
| Refresh | 1000 | 0.03 | 0 | 1e-4 | 0.3 | 0.5 | – | – | – | 1e-5 | 1 |
| Refresh | 2000 | 0.03 | 0 | 1e-4 | 0.3 | 0.5 | – | – | – | 1e-5 | 1 |
| STAR | 500 | 0.1 | 0 | 0 | – | – | 0.01 | 0.1 | 1 | – | – |
| STAR | 1000 | 0.1 | 0 | 0 | – | – | 0.05 | 0.1 | 1 | – | – |
| STAR | 2000 | 0.1 | 0 | 0 | – | – | 0.01 | 0.1 | 1 | – | – |

Table 8: **Average Accuracy on UCF101 and ActivityNet.** This table shows the average accuracy of iCaRL with and without IM across various sizes of memory buffer on two datasets (UCF101 and ActivityNet). The results demonstrate that the integration of information maximization (IM) regularizer consistently outperforms iCaRL on both datasets regardless of the memory setting.

| Dataset | UCF101 | | | ActivityNet | | |
|---|---|---|---|---|---|---|
| Buffer | 505 | 1010 | 2020 | 1000 | 2000 | 4000 |
| iCaRL | 68.44 | 74.70 | 81.07 | 41.72 | 45.05 | 46.91 |
| iCaRL (IM) | **75.60** | **79.11** | **83.53** | **45.76** | **50.01** | **55.20** |

our evaluation across multiple methods, datasets, and hyperparameter configurations, we employed a fixed random seed across all experiments to ensure fair comparison and computational feasibility.

To demonstrate the robustness of our findings, we provide results with multiple seeds (3 independent runs) for a representative configuration: DER with and without IM across all memory sizes on Split-CIFAR100 (Table 14).

The results demonstrate consistent improvements across all memory buffer sizes, with IM achieving gains of 7.53, 5.93, and 2.42 percentage points for buffer sizes of 500, 1000, and 2000, respectively. Importantly, the improvement margins substantially exceed the standard deviations, indicating that the observed gains are statistically reliable and not attributable to random variation.

Table 9: **Forgetting Rate on UCF101 and ActivityNet.** This table shows the forgetting rate of iCaRL with and without IM across various sizes of memory buffer on two datasets (UCF101 and ActivityNet). The results demonstrate that the integration of information maximization (IM) regularizer consistently achieves lower forgetting rates on both datasets regardless of the memory setting.

| Dataset | UCF101 | | | ActivityNet | | |
|---|---|---|---|---|---|---|
| Buffer | 505 | 1010 | 2020 | 1000 | 2000 | 4000 |
| iCaRL | 25.55 | 20.37 | 14.31 | 21.29 | 19.46 | 18.55 |
| iCaRL + IM | **16.87** | **16.02** | **10.54** | **19.01** | **18.55** | **15.48** |

Table 10: **Hyperparameter Sensitivity on ER (Memory=500).** This table shows the performance (Average Accuracy) of the ER baseline combined with different regularizers (IM, EM, EWC, SI) across three different values for the regularization strength $\lambda$. The best performance for each method is highlighted in bold.

| Method | $\lambda$=0.2 | $\lambda$=0.5 | $\lambda$=0.8 | Best $\lambda$ | Best Performance |
|---|---|---|---|---|---|
| ER (Baseline) | — | — | — | — | 21.6 |
| ER + IM | 24.46 | **33.6** | 24.06 | 0.5 | **33.6** |
| ER + EM | 22.16 | 20.8 | 16.32 | 0.2 | 22.16 |
| ER + EWC | 23.74 | 24.2 | **26.73** | 0.8 | 26.73 |
| ER + SI | 21.07 | **21.8** | 19.72 | 0.5 | 21.8 |

Table 11: **Hyperparameter Sensitivity on DER (Memory=500).** This table shows the performance (Average Accuracy) of the DER baseline combined with different regularizers (IM, EM, EWC, SI) across three different values for the regularization strength $\lambda$. The best performance for each method is highlighted in bold.

| Method | $\lambda$=0.2 | $\lambda$=0.5 | $\lambda$=0.8 | Best $\lambda$ | Best Performance |
|---|---|---|---|---|---|
| DER (Baseline) | — | — | — | — | 33.6 |
| DER + IM | 36.53 | **40.6** | 25.95 | 0.5 | **40.6** |
| DER + EM | **33.29** | 32.3 | 30.37 | 0.2 | 33.29 |
| DER + EWC | 19.09 | **20.5** | 13.89 | 0.5 | 20.5 |
| DER + SI | **32.8** | 29.5 | 28.07 | 0.2 | 32.8 |

Table 12: **Feature Drift Comparison Across Regularization Targets.** Feature drift is measured as mean L2 distance between class-specific feature representations after initial task learning versus after all tasks are completed. Results are reported on Split-CIFAR100 with Experience Replay baseline (buffer size 500).

| Task | Method | Mean Drift |
|------|--------|-----------|
| Task 1 | No IM (Baseline) | 21.21 |
| Task 1 | IM-CT | 23.80 |
| Task 1 | IM-BF | 24.26 |
| Task 2 | No IM (Baseline) | 9.74 |
| Task 2 | IM-CT | 12.01 |
| Task 2 | IM-BF | 11.20 |

Table 13: **Per-Task Accuracy After All Tasks Learned (Split-CIFAR100, ER with buffer size 500).** IM-CT substantially outperforms baseline on all old tasks (T1-T9) with expected trade-off on most recent task (T10).

| Task | No IM | IM-CT | IM-BF |
|------|-------|-------|-------|
| T1 | 8.5 | **18.8** | 9.3 |
| T2 | 7.1 | **22.7** | 6.7 |
| T3 | 11.4 | **34.8** | 12.6 |
| T4 | 6.4 | **19.0** | 7.7 |
| T5 | 18.6 | **35.0** | 21.0 |
| T6 | 13.2 | **32.1** | 10.2 |
| T7 | 15.2 | **29.0** | 16.7 |
| T8 | 19.5 | **32.1** | 20.5 |
| T9 | 23.9 | **41.4** | 26.6 |
| T10 | 92.0 | 77.2 | **93.2** |
| **Mean (T1-T9)** | 13.8 | **29.4** | 14.6 |

Table 14: **Multiple Runs Analysis: DER on Split-CIFAR100.** Results report mean accuracy and standard deviation across 3 independent runs. Improvement margins consistently exceed standard deviations, indicating statistical reliability.

| Memory Size | DER (mean $\pm$ std) | DER + IM (mean $\pm$ std) | Improvement |
|-------------|---------------------|---------------------------|-------------|
| 500 | $33.57 \pm 1.42$ | $41.10 \pm 0.55$ | +7.53 |
| 1000 | $43.35 \pm 0.68$ | $49.28 \pm 0.52$ | +5.93 |
| 2000 | $51.46 \pm 0.52$ | $53.89 \pm 0.71$ | +2.42 |

