# OpenReview forum: "Forget Less, Retain More: A Lightweight Regularizer for Rehearsal-Based Continual Learning"
_TMLR — Accepted by TMLR_

### Review · Reviewer_dHxS · 2025-09-22

**Summary Of Contributions:**

This paper explores continual learning by introducing a regularizer into rehearsal-based methods. Specifically, it proposes an information maximization regularizer applied to the model outputs to encourage learning new tasks while mitigating forgetting of previous ones. The regularizer comprises two components: entropy minimization and diversity maximization. Extensive experiments on both image- and video-based continual learning benchmarks demonstrate the effectiveness of the proposed approach.

**Additional Comments:**

See above.

**Audience:**

No

**Audience Explanation:**

The proposed solution lacks novelty, and the results are neither surprising nor state-of-the-art.

**Broader Impact Concerns:**

N.A.

**Claims And Evidence:**

Yes

**Claims Explanation:**

The main contributions of this paper lie in its evaluation, which is supported by substantial experimental results.

**Requested Changes:**

1. This paper appears to be a special case of the prior work [1], sharing a similar evaluation protocol and writing style, but it does not provide a direct comparison with that work.
2. The motivation behind $L_{div}$ is insufficiently explained. It seems to primarily serve as a form of prototype learning.
3. The set of baselines is too limited; more recent and representative baselines should be included for a fair comparison [2][3].
4. The three claimed contributions could be consolidated into a single, unified contribution for clarity.

References:

[1] A unified and general framework for continual learning. ICLR 2024.

[2] Continual learning with tiny episodic memories. https://arxiv.org/abs/1902.10486.

[3] Dark experience for general continual learning: a strong, simple baseline. NeurIPS 2020.

---

> ### Author Response · Authors · 2025-11-20
> **Response to: “The three claimed contributions could be consolidated…”**
>
> We thank the reviewer for this constructive suggestion. We agree with this suggestion and have revised our contribution statement from three to two main contributions. Specifically, we merged the first two contributions (systematic evaluation of IM and its orthogonality to design choices) into a single unified contribution. We retained the video continual learning extension as a separate contribution, as it represents a non-trivial generalization to a different data modality with additional complexity.
> This revised structure improves clarity, while preserving the significance of extending to a new domain. These changes will be reflected in the revised manuscript.

---

> ### Author Response · Authors · 2025-11-20
> **Response to: “The set of baselines is too limited…”**
>
> The reviewer's suggested baselines are already included in our evaluation. Regarding the reviewer's reference  [3] "Dark experience for general continual learning", is already included as DER and DER++ (Figure 2 and Tables 4-5).
> Regarding the reviewer's reference [2] "On Tiny Episodic Memories in Continual Learning", this is already included as ER (Figure 2 and Tables 4-5). The main contribution of [2] is demonstrating that a simple Experience Replay (ER) baseline, even when using a very tiny episodic memory, can outperform many specifically designed continual learning approaches.
>
> We respectfully disagree with the reviewer regarding the limitation of baselines, our evaluation includes methods published between 2019 and 2025 (covering both foundational and recent approaches in continual learning):
>
> - 2019: Experience Replay (ER) [4]
> - 2020: Dark Experience Replay (DER) and DER++ [3]
> - 2024: Refresh Learning [1]
> - 2025: STAR [5]
>
> **References**
>
> [1] Zhenyi Wang, Yan Li, Li Shen, and Heng Huang. A unified and general framework for continual learning. In International Conference on Learning Representations (ICLR), 2024.
>
> [2] Arslan Chaudhry, Marcus Rohrbach, Mohamed Elhoseiny, Thalaiyasingam Ajanthan, Puneet K. Dokania, Philip H. S. Torr, and Marc'Aurelio Ranzato. On tiny episodic memories in continual learning. arXiv preprint arXiv:1902.10486, 2019.
>
> [3] Pietro Buzzega, Matteo Boschini, Angelo Porrello, Davide Abati, and Simone Calderara. Dark experience for general continual learning: a strong, simple baseline. In Advances in Neural Information Processing Systems (NeurIPS), volume 33, pages 15920–15930, 2020.
>
> [4] David Rolnick, Arun Ahuja, Jonathan Schwarz, Timothy Lillicrap, and Gregory Wayne. Experience replay for continual learning. Advances in Neural Information Processing Systems, 32, 2019.
>
> [5] Masih Eskandar, Tooba Imtiaz, Davin Hill, Zifeng Wang, and Jennifer Dy. STAR: Stability-inducing weight perturbation for continual learning. arXiv preprint arXiv:2503.01595, 2025.

---

> ### Author Response · Authors · 2025-11-20
> **Response to: “The motivation behind L_div is insufficiently explained...”**
>
> We thank the reviewer for this insightful comment. We agree that a clearer explanation of the motivation behind the L_div term will enhance the paper, and we have expanded on it below.
>
> The $L_{div}$ loss is grounded in the principle of mutual information maximization between inputs X and predicted outputs Y [2], formalized as: I(X;Y) = H(Y) − H(Y|X)
>
> This decomposition reveals two complementary goals:
> 1. Minimize conditional entropy H(Y∣X): ensures the model makes confident predictions for each input.
> 2. Maximize marginal entropy H(Y): ensures the predictions are distributed across all output classes, avoiding collapse to a subset.
>
> Following this principle, SHOT [1] implements mutual information maximization [3, 4] via two loss terms (which we adopted for our IM regularizer):
> - $L_{ent}$ minimizes H(Y∣X)
> - $L_{div}$ maximizes H(Y)
>
> The $L_{div}$ term specifically addresses the second objective: promoting diversity in predictions across the batch. Without this term, the model could trivially minimize $L_{ent}$ by assigning high-confidence predictions to a single class for all inputs. $L_{div}$ prevents this by encouraging the model’s average predictions to span the label space.
>
> While $L_{div}$ may appear superficially similar to prototype learning, its motivation and mechanism are fundamentally different. Prototype-based methods such as iCaRL [5] and CoPE [6] rely on storing class-wise feature prototypes (typically mean representations of each class in feature space). These prototypes are used during inference for classification via nearest-prototype matching, and require additional memory proportional to the number of classes and feature dimension.
>
> In contrast, $L_{div}$ operates entirely during training, and does not store or compute prototypes. Instead, it regularizes the model’s output distribution by maximizing the marginal entropy of predictions. This encourages diversity across predicted classes within a batch and mitigates the risk of prediction collapse. We will incorporate this expanded explanation into the revised version of the manuscript to improve clarity and completeness.
>
> **References**
>
> [1] Jian Liang, Dapeng Hu, and Jiashi Feng. Do we really need to access the source data? Source hypothesis transfer for unsupervised domain adaptation. In ICML, pages 6028–6039, 2020.
>
> [2] Claude E. Shannon. A mathematical theory of communication. Bell System Technical Journal, 27(3):379–423, 1948.
>
> [3] Andreas Krause, Pietro Perona, and Ryan G Gomes. Discriminative clustering by regularized information maximization. In NeurIPS, 2010.
>
> [4] Weihua Hu, Takeru Miyato, Seiya Tokui, Eiichi Matsumoto, and Masashi Sugiyama. Learning discrete representations via information maximizing self-augmented training. In ICML, pages 1558–1567, 2017.
>
> [5] Sylvestre-Alvise Rebuffi, Alexander Kolesnikov, Georg Sperl, and Christoph H Lampert. iCaRL: Incremental classifier and representation learning. In CVPR, pages 2001–2010, 2017.
>
> [6] Aristotelis Chrysakis and Marie-Francine Moens. Online continual learning from imbalanced data. In ICML, pages 1952–1961, 2020.

---

> ### Author Response · Authors · 2025-11-20
> **Response to: “This paper appears to be a special case of the prior work [1]...”**
>
> We respectfully disagree with this assessment. We clarify that our paper has no overlapping contributions with [1]. Moreover, our empirical analysis shows that the unlearning mechanism of [1] and our proposed regularization scheme are in fact complementary producing improved results when they are paired together.
>
>
> **No Overlapping contribution**
>
> The related work in [1] makes two contributions: (1) a general theoretical framework that unifies existing continual learning methods via Bregman divergence, and (2) the Refresh Learning approach, which introduces an unlearn–relearn mechanism. Our paper does not overlap with any of these claims. We don't claim a generalized continual learning framework, instead, we focus on analyzing the effectiveness of information maximization as a regularization strategy for continual learning. Empirically, we demonstrate that this regularization scheme is effective for several current CL methods across the image and video domains. We also note that [1] does not include any claims or experiments in the video domain.
>
>
> **Similar Evaluation Protocol**
>
> Our paper follows a similar evaluation protocol to [1] as this is the standard setup for benchmarking CL methods. Split-Cifar100 Split-Tiny ImageNet are ubiquitous evaluation benchmarks which are found in most CL papers. We select these datasets not only to provide standard evaluation benchmarks, but also to provide direct and fair comparison points. Similarly, we choose a similar group of CL methods as [1] since these are the most representative CL approaches.  Our evaluation protocol is deliberately similar to [1], and many other papers which streamlines the evaluation and review process.
>
>
> **Comparison to Refresh Learning**
>
> We view our approach as fundamentally distinct from Refresh Learning. Refresh Learning operates partly in weight space, guiding parameter update trajectories through the unlearn–relearn steps using the Fisher Information Matrix. In contrast, our method is agnostic to the weight space and does not explicitly shape the trajectory or magnitude of weight updates. Instead, we modulate the magnitude of the loss through a self-supervised information-maximization objective, which does not depend on the current weights or on individual gradients propagated through the network.
>
> When Refresh Learning operates in output space, it relies on a reference vector (labeled as z in the original paper [1]) to help preserve knowledge from previously learned tasks. In contrast, our information-maximization objective is a fully self-supervised loss that does not require any reference instantiation from prior tasks (such as the vector z) nor any estimation of the Bregman divergence. IM operates exclusively on batch-level statistics, which remain entirely independent of past tasks and past network outputs.
>
> Finally, our manuscript provides a direct comparison with Refresh Learning [1]. Our empirical evaluation includes Refresh Learning as a baseline method, Figure 2 and Tables 4-5 outline results for both "Refresh" (Refresh Learning [1]) and "Refresh+IM" (our method applied to Refresh Learning). We note that our approach (IM+Refresh) yields an average accuracy improvement of 1-2%  and 2-8% on Split-CIFAR100 and Split-Tiny ImageNet, respectively, compared to Refresh alone. These results empirically support that our proposal is not a special case of Refresh Learning, rather a different, and complementary method for CL whose performance can go beyond the proposal of [1].
>
> We see no direct relationship or correspondence between our method and [1]. Moreover, the complementary nature strongly suggests that these are orthogonal approaches that can be independently used.
>
> **References**
>
> [1] Zhenyi Wang, Yan Li, Li Shen, and Heng Huang. A unified and general framework for continual learning. In International Conference on Learning Representations (ICLR), 2024.

---

### Review · Reviewer_6WE2 · 2025-10-18

**Summary Of Contributions:**

The topic of improving continual learning performance is important and broadly relevant. The proposed approach is clearly described (adding additional regularization terms in the training loss), and the simulation results indicate that it achieves higher accuracy and reduced forgetting.

However, a key concern is the lack of justification for the proposed methodology. In particular, the introduction of the term $L_{\text{div}}$ in Eq. (2) is not well motivated. In fact, $L_{\text{div}}$ is the core part of the proposed approach. It is unclear why this term is beneficial, or why it appears with a positive sign rather than a negative one like $L_{\text{ent}}$. As written, $L_{\text{div}}$ seems ad hoc and its role is confusing.

**Audience:**

Yes

**Audience Explanation:**

This paper proposed a new approach to improve the performance of continual learning. Researchers in the field of continual learning will be interested.

**Claims And Evidence:**

Yes

**Claims Explanation:**

There are many simulation results to illustrate the performance of the proposed approach.

**Requested Changes:**

1. In Eq. (2), the meanings of the subscripts "ent" and "div" are not explained. Please clarify what these terms denote.

2. Based on the definition of $\hat{f}\_{\theta}(x)$ as the expectation of $f\_{\theta}(x)$ over $x$, the expression should no longer depend on $x$. I recommend removing $(x)$ from the notation $\hat{f}\_{\theta}(x)$.

3. In the definition of $R\_{\text{EWC}}$, the roles of the indices $i$ and $t$ are unclear. Additionally, since $F$ is a matrix, it is ambiguous whether $F\_i$ refers to a row, a column, or some other component. This should be specified.

4. In the definition of $R\_{\text{SI}}$, the summation index is written as $\sum\_t^T$; should this instead be $\sum\_{t=1}^T$? Also, the quantity $\omega\_t^k$ is not defined.

5. It appears that $L\_{\text{ent}}$ and $R\_{\text{EM}}$ are identical. If that is the case, the notation should be made consistent.

6. All reported simulation results are given as single numbers. Are these results averaged over multiple runs? Reporting the variance or standard deviation would make the evaluation more convincing.

---

> ### Author Response · Authors · 2025-11-22
> **Response to: “ In Eq. (2), the meanings of the subscripts "ent" and "div" ..”**
>
> We thank the reviewer for pointing this out. The $L_{ent}$ and $L_{div}$ losses are grounded in the principle of mutual information maximization between inputs X and predicted outputs Y [2], formalized as: I(X;Y) = H(Y) − H(Y|X)
> This decomposition reveals two complementary goals:
> 1. Minimize conditional entropy H(Y∣X): ensures the model makes confident predictions for each input.
> 2. Maximize marginal entropy H(Y): ensures the predictions are distributed across all output classes, avoiding collapse to a subset.
>
> Following this principle, SHOT [1] implements mutual information maximization [3] [4] via two loss terms, which we adopted for our IM regularizer:
> - $L_{ent}$ minimizes H(Y∣X)
> - $L_{div}$ maximizes H(Y)
>
> The subscripts denote:
> - "ent" stands for entropy (entropy minimization term)
> - "div" stands for diversity (diversity maximization term)
>
> **References**
>
> [1] Jian Liang, Dapeng Hu, and Jiashi Feng. Do we really need to access the source data? Source hypothesis transfer for unsupervised domain adaptation. In ICML, pages 6028–6039, 2020.
> [2] Claude E. Shannon. A mathematical theory of communication. Bell System Technical Journal, 27(3):379–423, 1948.
>
> [3] Andreas Krause, Pietro Perona, and Ryan G Gomes. Discriminative clustering by regularized information maximization. In NeurIPS, 2010.
>
> [4] Weihua Hu, Takeru Miyato, Seiya Tokui, Eiichi Matsumoto, and Masashi Sugiyama. Learning discrete representations via information maximizing self-augmented training. In ICML, pages 1558–1567, 2017.

---

> ### Author Response · Authors · 2025-11-22
> **Response to mathematical notation questions**
>
> We thank the reviewer for these careful observations regarding mathematical notation.
>
>
> **Notation f̂_θ(x)**
>
> Reviewer suggests that since f̂_θ(x) is defined as the expectation of f_θ(x) over x, it should not depend on x. we agree with that and will modify the notation to f̂_θ (removing the dependence on x) throughout the manuscript.
>
> **Indices in R_EWC definition**
>
> We agree with the reviewer that the roles of indices i and t, as well as the meaning of $F_{i}$, need clarification. In the EWC formulation:
>
> - t indexes tasks (t = 1, ..., T)
> - i indexes parameters ($θ_{i}$ is the i-th parameter)
> - F is the Fisher Information Matrix, where $F_{i}$ refers to the i-th diagonal element (corresponding to parameter $θ_{i}$)
>
> We will include all these clarifications in the revised manuscript.
>
> **R_SI definition**
>
> We agree with the reviewer's point that the summation index should be $Σ_{t=1}^T$ (not $Σ_{t}^T$).
>
> The quantity $ω_t^k$ represents the importance weight for parameter k at task t.
>
> We will integrate these changes and clarifications in the revised manuscript.
>
> **$L_{ent}$ and $R_{EM}$ notation**
>
> We agree with the reviewer's point that $L_{ent}$ (our entropy minimization term) and $R_{EM}$ (Entropy Minimization regularizer from prior work) serve the same purpose (are identical). For the revised manuscript we will clarify this and make the notations consistent as suggested by the reviewer.

---

> ### Author Response · Authors · 2025-11-22
> **Response to: “All reported simulation results are given as single numbers ..”**
>
> Our experimental methodology involved:
>
> 1- Initial validation: We conducted preliminary experiments with multiple seeds to validate the observed behaviors and trends
> 2- Fixed seed protocol: Given the extensive scope of our evaluation, we used a fixed random seed across all experiments to ensure fair comparison and computational feasibility
>
> To demonstrate the robustness of our findings, we provide results (average accuracy) with multiple seeds (3 runs) for DER with and without IM across all memory sizes on Split-CIFAR100 (in the table below).
>
> | Memory Size | DER (mean ± std) | DER + IM (mean ± std) | Improvement |
> |:-----------:|:----------------:|:---------------------:|:-----------:|
> | 500         | 33.57 ± 1.42     | 41.10 ± 0.55          | +7.53       |
> | 1000        | 43.35 ± 0.68     | 49.28 ± 0.52          | +5.93       |
> | 2000        | 51.46 ± 0.52     | 53.89 ± 0.71          | +2.42       |
>
>
> We observe consistent improvements, where IM shows consistent gains across all memory settings. The improvement margins exceed the standard deviations, demonstrating the reliability of our findings. We will add these results with provided clarifications to the revised manuscript.

---

### Review · Reviewer_Ebdg · 2025-11-09

**Summary Of Contributions:**

The paper proposes a regularization-based method for continue learning named information maximization. Compared to classical rehearsal-based approaches, it add two additional objectives: 1) an entropy loss to encourage confident predictions; 2) a diverse loss to promote diverse predictions across the batch. The method is validated in CIFAR-100 and Tiny ImageNet, demonstrating improved accuracy and less forgetting over baseline methods.

**Audience:**

Yes

**Audience Explanation:**

The researchers in the community might find this method interesting as it is relatively light-weight compared to the traditional memory-based methods.

**Broader Impact Concerns:**

N/A.

**Claims And Evidence:**

Yes

**Claims Explanation:**

While most claims are accurate, there are some missing experiments that make the results unconvincing:
* The paper does not detail the tuning process for the critical hyperparameters (e.g., the $\lambda$ for EWC or SI) for these competing methods; the claim that IM is superior to other regularization strategies is not fully supported.
* The authors find that applying the IM regularizer only to the current task (CT) samples yields the best performance (Table 3). This is counter-intuitive for a method designed to reduce forgetting of old tasks. One would logically expect the most effective regularization to be applied to the rehearsal buffer (BF) samples, which directly represent the old knowledge.

**Requested Changes:**

* **Fair hyperparameter search for baseline regularizers.** The current claim that IM is superior to EWC, SI, and EM is not convincingly supported. The performance of these baseline regularizers, particularly EWC, is anomalously poor when combined with rehearsal methods (e.g., dropping DER/DER++ accuracy by nearly half). The authors shall conduct and report a fair hyperparameter search for the competing regularizers (EWC, SI, EM).
* **More comprehensive explain of the "current task (CT) only" result.** The ablation study in Table 3 finds that applying the IM regularizer exclusively to the current task samples (CT) yields better performance than applying it to the buffer (BF) or both (ALL). The authors shall add a new analysis section to investigate why CT-only regularization is the most effective strategy for preventing forgetting, which should be beyond reporting accuracy and explore the underlying dynamics. For instance, authors may analyze the feature representations to see if CT regularization prevents feature drift or collapse for old classes stored in the buffer.

---

> ### Author Response · Authors · 2025-11-22
> **Response to Hyperparameter Fairness Concern**
>
> We thank the reviewer for raising this important concern about fair hyperparameter comparison. We address this by: (1) explaining our evaluation methodology, (2) presenting a sensitivity analysis for our main results with regularizers and (3) demonstrating the robustness of our findings.
>
> **Evaluation Methodology and Practical Considerations**
>
> Conducting exhaustive hyperparameter search for every method across all experimental conditions would require an infeasible amount of experiments. Critically, this approach has limited practical value:
>
> 1- Real-world deployment: Practitioners typically cannot afford extensive hyperparameter tuning for every new scenario or dataset.
>
> 2- Method comparison: If each method requires different optimal hyperparameters for different settings, it becomes unclear which method is genuinely superior versus which is easier to tune.
>
>
> We adopted a fixed hyperparameter protocol to evaluate methods under realistic conditions:
>
> 1- Consistent λ=0.5 for all regularizers (IM, EM, EWC, SI): Eliminates method-specific hyperparameter advantages and tests robustness.
>
> 2- Standard configurations: For EWC and SI, we follow the configurations outlined in the original papers and in the widely used Mammoth continual learning framework.
>
> 3- Fair experimental conditions: All methods evaluated under the same base method, memory size, dataset, and training protocol.
>
> **Sensitivity Analysis**
>
> To validate that our findings are not artifacts of the specific λ=0.5 choice, we conducted sensitivity analysis across λ ∈ {0.2, 0.5, 0.8} on Split-CIFAR100 (Table1 and Table 2).
>
> **Table 1: Hyperparameter Sensitivity on ER (Memory=500)**
>
> | Method      | λ=0.2 | λ=0.5 | λ=0.8 | Best λ | Best Performance |
> |:-----------:|:-----:|:-----:|:-----:|:------:|:----------------:|
> | ER          | -     | -     | -     | -      | 21.6             |
> | ER + IM     | 24.46 | 33.6  | 24.06 | 0.5    | 33.6             |
> | ER + EM     | 22.16 | 20.8  | 16.32 | 0.2    | 22.16            |
> | ER + EWC    | 23.74 | 24.2  | 26.73 | 0.8    | 26.73            |
> | ER + SI     | 21.07 | 21.8  | 19.72 | 0.5    | 21.8             |
>
> **Table 2: Hyperparameter Sensitivity on DER (Memory=500)**
>
>
> | Method      | λ=0.2 | λ=0.5 | λ=0.8 | Best λ | Best Performance |
> |:-----------:|:-----:|:-----:|:-----:|:------:|:----------------:|
> | DER         | -     | -     | -     | -      | 33.6             |
> | DER + IM    | 36.53 | 40.6  | 25.95 | 0.5    | 40.6             |
> | DER + EM    | 33.29 | 32.3  | 30.37 | 0.2    | 33.29            |
> | DER + EWC   | 19.09 | 20.5  | 13.89 | 0.5    | 20.5             |
> | DER + SI    | 32.8  | 29.5  | 28.07 | 0.2    | 32.8             |
>
>
> **Robustness of performance ranking**
>
> Our proposal outperforms every other regularizer, even when each method uses its best λ value from the sensitivity analysis:
>
> - ER baseline: 21.6
> - ER + IM (λ=0.5): 33.6 → **+12.0 improvement**
> - ER + EWC (λ=0.8): 26.73 → +5.1 improvement
> - ER + EM (λ=0.2): 22.16 → +0.6 improvement
> - ER + SI (λ=0.5): 21.8 → +0.2 improvement
>
> - DER baseline: 33.6
> - DER + IM (λ=0.5): 40.6 → **+7.0 improvement**
> - DER + SI (λ=0.2): 32.8 → -0.8 degradation
> - DER + EM (λ=0.2): 33.29 → -0.3 degradation
> - DER + EWC (λ=0.5): 20.5 → -13.1 degradation
>
>
> The performance ranking remains consistent regardless of whether we use:
> - Fixed λ=0.5 for all methods (our main results)
> - Individually optimal λ for each method (sensitivity analysis)
>
> This demonstrates that our conclusions are robust to hyperparameter choices and not artifacts of favoring IM through hyperparameter selection.
>
>
> **Regarding anomalous EWC/SI performance**
>
> The reviewer correctly notes that EWC's performance appears anomalously poor (e.g., dropping DER from 33.6 to 20.5). Our sensitivity analysis shows this is not due to suboptimal λ selection:
> - EWC on DER achieves 19.09, 20.5, and 13.89 across λ ∈ {0.2, 0.5, 0.8}—all represent substantial degradation.
> - SI similarly shows minimal improvement or degradation across all tested λ values on DER.
>
> We will include the sensitive analysis and needed clarification to the revised manuscript.

---

### Author Response · Authors · 2025-11-22
**Response to CT-Only Ablation Analysis Request (1/2)**

We thank the reviewer for this insightful suggestion. We have conducted a comprehensive feature drift analysis to investigate the underlying mechanism of why CT-only regularization outperforms buffer regularization. Our analysis reveals a counterintuitive but significant finding: effective continual learning regularization does not necessarily minimize feature drift; instead, it can enable controlled feature evolution that maintains class discriminability.


### **Feature Drift Analysis**

We measured feature drift as the mean L2 distance between class-specific feature representations (penultimate layer activations) immediately after learning a task versus after all tasks are completed. We analyzed both Task 0 (classes 0-9) and Task 1 (classes 10-19) to demonstrate consistency across tasks.

**Counterintuitive Finding: Higher Drift, Better Performance**

Table1 shows feature drift comparison across both tasks. Contrary to the intuition that lower drift indicates better performance, we observe that IM-CT increases feature drift more than the baseline:
- Task 0: IM-CT increases drift by +12.2% (21.21 → 23.80)
- Task 1: IM-CT increases drift by +23.3% (9.74 → 12.01)

Interestingly, IM-BF also increases drift, sometimes even exceeding IM-CT (Task 0: 24.26 vs 23.80). This raises a critical question: if both methods increase drift, why does only IM-CT improve performance?



**Table 1: Feature Drift Comparison (Task 0 and Task 1).** Feature drift measured as mean L2 distance between class features after initial learning vs. after all tasks. Both Task 0 and Task 1 show that IM-CT increases drift compared to the baseline.
| Task   | Method     | Mean Drift | Drift Increase |
|:------:|:----------:|:----------:|:--------------:|
| Task 0 | No IM      | 21.21      | —              |
| Task 0 | IM-CT      | 23.80      | +12.2%         |
| Task 0 | IM-BF      | 24.26      | +14.4%         |
|        |            |            |                |
| Task 1 | No IM      | 9.74       | —              |
| Task 1 | IM-CT      | 12.01      | +23.3%         |
| Task 1 | IM-BF      | 11.20      | +15.0%         |


**Drift Does Not Equal Forgetting**

Table 2 reveals the answer: despite higher feature drift, IM-CT dramatically improves old task accuracy, while IM-BF provides minimal or negative benefit:
Old Task Performance (Tasks 0-8 mean accuracy):
- No IM: 13.8%
- IM-CT: 29.4% (+15.6 points, +114% improvement)
- IM-BF: 14.6% (+0.8 points, +6% improvement)
This demonstrates that drift magnitude alone does not determine performance. The key is where and how regularization is applied.

**Table 2: Old Task Accuracy Comparison.** Mean accuracy on old tasks (Tasks 0-8) after all tasks are learned.

| Method     | Old Tasks Accuracy (T0-T8) | Improvement vs Baseline |
|:----------:|:--------------------------:|:-----------------------:|
| No IM      | 13.8%                      | —                       |
| IM-CT      | 29.4%                      | +15.6% (+114%)          |
| IM-BF      | 14.6%                      | +0.8% (+6%)             |

Evidence from per-task breakdown (Table 3) that IM-CT substantially outperforms baseline across all old tasks (T0-T8), with individual task accuracies ranging from 18.8% to 41.4%, compared to 6.4% to 23.9% for baseline. This consistent improvement across tasks indicates systematic feature refinement, not random drift.

**Table 3: Per-Task Accuracy Breakdown.** Per-task accuracy after all tasks are learned (Split-CIFAR100, ER with buffer=500). IM-CT substantially outperforms baseline on all old tasks (T0-T8) with expected trade-off on most recent task (T9).

| Task | No IM | IM-CT | IM-BF | IM-CT Improvement |
|:----:|:-----:|:-----:|:-----:|:-----------------:|
| T0   | 8.5   | 18.8  | 9.3   | +10.3 (+121%)     |
| T1   | 7.1   | 22.7  | 6.7   | +15.6 (+220%)     |
| T2   | 11.4  | 34.8  | 12.6  | +23.4 (+205%)     |
| T3   | 6.4   | 19.0  | 7.7   | +12.6 (+197%)     |
| T4   | 18.6  | 35.0  | 21.0  | +16.4 (+88%)      |
| T5   | 13.2  | 32.1  | 10.2  | +18.9 (+143%)     |
| T6   | 15.2  | 29.0  | 16.7  | +13.8 (+91%)      |
| T7   | 19.5  | 32.1  | 20.5  | +12.6 (+65%)      |
| T8   | 23.9  | 41.4  | 26.6  | +17.5 (+73%)      |
| T9   | 92.0  | 77.2  | 93.2  | -14.8 (-16%)      |
|      |       |       |       |                   |
| **Mean (T0-T8)** | **13.8** | **29.4** | **14.6** | **+15.6 (+114%)** |

---

> ### Author Response · Authors · 2025-11-22
> **Response to CT-Only Ablation Analysis Request (2/2)**
>
> Despite increasing feature drift to similar or higher levels than IM-CT (Task 0: 24.26 vs 23.80 - Table 1), IM-BF provides minimal accuracy improvement. The empirical evidence shows:
>
> - Limited benefit on old tasks: IM-BF achieves nearly identical old task performance to baseline (14.6% vs 13.8% - Table 2), representing only a +6% improvement compared to IM-CT's +114% improvement.
>
> - On Task 1 specifically, IM-BF actually degrades performance (6.7% vs 7.1% baseline - Table 3), while on Task 0 it shows modest improvement (9.3% vs 8.5% baseline - Table 3).
>
> While we cannot definitively determine the exact mechanism, the data suggests that applying regularization to buffer samples may not provide the same benefits as applying it to current task samples. One hypothesis is that the buffer already provides direct supervision for old classes through data replay, potentially making additional regularization on these samples less effective or even counterproductive.
>
> ### **Understanding the Trade-off**
>
> Table 3 shows the per-task accuracy breakdown, revealing an expected and desirable trade-off: IM-CT sacrifices some performance on the most recent task (T9: 92.0% → 77.2%, -14.8%) to achieve substantial gains on old tasks (mean: 13.8% → 29.4%, +15.6%).
> This trade-off is characteristic of effective continual learning: the model balances stability (retaining old knowledge) and plasticity (learning new knowledge). In contrast, IM-BF maintains high performance on T9 (93.2%) but fails to prevent forgetting on old tasks, indicating it has not addressed the core challenge of continual learning.
>
> We will include the provided analysis to the revised manuscript.

---

### Decision · Action_Editor_WcMv · 2025-12-19

**Recommendation:** Accept with minor revision

**Additional Comments:**

Please add the following in the revision (among other suggestions by the reviewers)
1. The justification of the regularizer given to the reviewers, e.g., https://openreview.net/forum?id=CJw1ZjkJMG&noteId=uvFvI0QDES
2. include the sensitivity analysis results (Tables 1 and 2 from the rebuttal) and the associated discussion on robustness in the final manuscript as promised.

Sometimes memoty M_t is written with \mathcal and sometime without. Please fix.

**Audience:**

Yes

**Audience Explanation:**

This paper addresses the problem of continual learning which is highly relevant for TMLR's audience.

**Claims And Evidence:**

Yes

**Claims Explanation:**

This paper proposes a new Information Maximization regularizer for continual learning. The regularizer encourages confident and diverse predictions on a set of examples (X) which is easy to compute. I am a bit unsure why the two do not conflict, but the author give an explanation based on mutual information based on existing work. The reviewer seem to be satisfied with this, although all of the believe that the work is of incremental nature. Nevertheless, they all agree to accept it because the work could be useful for the community to improve things.